# Vat Photopolymerization 3D-Printing of Dynamic Thiol-Acrylate Photopolymers Using Bio-Derived Building Blocks

**DOI:** 10.3390/polym14245377

**Published:** 2022-12-08

**Authors:** Usman Shaukat, Bernhard Sölle, Elisabeth Rossegger, Sravendra Rana, Sandra Schlögl

**Affiliations:** 1Polymer Competence Center Leoben GmbH, Roseggerstrasse 12, 8700 Leoben, Austria; 2School of Engineering, Energy Acres, University of Petroleum & Energy Studies (UPES), Dehradun 248007, India

**Keywords:** dynamic polymer networks, vitrimers, DLP 3D-printing, self-healing, bio-based monomers, photopolymers, shape-memory

## Abstract

As an energy-efficient additive manufacturing process, vat photopolymerization 3D-printing has become a convenient technology to fabricate functional devices with high resolution and freedom in design. However, due to their permanently crosslinked network structure, photopolymers are not easily reprocessed or repaired. To improve the environmental footprint of 3D-printed objects, herein, we combine the dynamic nature of hydroxyl ester links, undergoing a catalyzed transesterification at elevated temperature, with an acrylate monomer derived from renewable resources. As a sustainable building block, we synthesized an acrylated linseed oil and mixed it with selected thiol crosslinkers. By careful selection of the transesterification catalyst, we obtained dynamic thiol-acrylate resins with a high cure rate and decent storage stability, which enabled the digital light processing (DLP) 3D-printing of objects with a structure size of 550 µm. Owing to their dynamic covalent bonds, the thiol-acrylate networks were able to relax 63% of their initial stress within 22 min at 180 °C and showed enhanced toughness after thermal annealing. We exploited the thermo-activated reflow of the dynamic networks to heal and re-shape the 3D-printed objects. The dynamic thiol-acrylate photopolymers also demonstrated promising healing, shape memory, and re-shaping properties, thus offering great potential for various industrial fields such as soft robotics and electronics.

## 1. Introduction

Additive manufacturing (AM) or 3D-printing is gaining increasing attention as a versatile processing technique for designing polymeric prototypes [1,2,3,4,5]. To date, 3D-printed polymers have been successfully integrated in electronics [6,7], automotive [8], robotics [7,9], tissue engineering, and biomedical industries [10,11,12,13,14,15,16,17]. With the rising development of new raw materials and technological innovations for 3D-printing, the global market potential of AM technologies has grown exponentially over the past years and is projected to reach approximately $20 billion by 2029 [18,19]. Using vat photopolymerization 3D-printing for the fabrication of 3D-objects offers numerous advantages, including ease of raw materials’ handling, ambient printing conditions, and the ability to form complex structures at a reasonable build time with high resolution. However, a major drawback of photopolymers is their lack in recyclability and repairability, as liquid resin transforms to a solid covalently-crosslinked polymer network during the printing process.

The introduction of dynamic covalent bonds is one way to render 3D-printed photopolymers repairable and reusable. In particular, dynamic polymer networks following an associative bond exchange mechanism exhibit excellent thermal and mechanical properties, and behave like classic thermosets below their topology freezing temperature (T_v_). Increasing the temperature above the T_v_ promotes a rearrangement of the network’s topology, which induces a macroscopic viscoelastic reflow [20,21]. This material flow follows the Arrhenius trend and can be exploited for the reprocessing, healing, or welding of covalently-crosslinked polymer networks [22,23,24]. Numerous dynamic bond exchange mechanisms have been studied [23,25] over the past years and successfully applied in photopolymer networks. In particular, Zhang and co-workers developed a photo-curable acrylate resin containing dynamic hydroxy ester moieties. By exploiting thermo-activated transesterification, the authors were able to print healable and reprocessable photopolymer networks [24]. Following the same material concept, Li et al. recently demonstrated the recycling of 3D-printed structures [25]. However, in both studies, Zn(OAc)_2_ was applied as a transesterification catalyst, which is insoluble in the majority of common acrylate monomers and thus, limits the versatility of network design.

In previous work, we have demonstrated the use of organic phosphate esters as efficient transesterification catalysts in 3D-printable vitrimers [26,27,28,29]. Compared to other catalysts, they are liquid, easily soluble in various monomers, covalently incorporated into the network across their methacrylate moieties, and do not compromise the curing rate of radically-induced photoreactions. We also investigated the effect of the pK_a_ value (negative logarithm of the acid dissociation constant (K_a_) in solution) of phosphate catalysts on the catalytic efficiency of transesterification reactions and found that an increasing acidity increased the bond exchange rate [28]. 

Along with recyclability and repairability, research in 3D-printing is also geared towards the synthesis of functional and bio-based monomers derived from plant/vegetable oils, rosin, or carbohydrates as feedstocks [30,31,32,33,34]. Plant and vegetable oils have unsaturated triglyceride structures and are abundantly available, biodegradable, and environmentally friendly [30,35,36,37,38,39,40,41]. The oxidation of the unsaturated glyceride moieties leads to the formation of epoxide groups, which can be converted to acrylate derivatives by nucleophilic ring opening with acrylic acid [30,40,42,43,44]. Combining dynamic polymer networks with bio-based resins can further reduce the overall environmental footprint of photocurable systems. Fei et al. previously demonstrated the successful synthesis of UV-curable dimethacrylate compounds from a vegetable oil-derived dimer acid and achieved rapid transesterification reactions at elevated temperatures by using an internal tertiary amine as a catalyst. The DLP-printed objects offered excellent welding and shape-changing properties [45]. Cortés-Guzmán and co-workers focused on the development of vanillin-derived methacrylate resins for 3D-printing, which undergo catalyst-free transimination simply through the application of heat [46]. Recently, Zhu et al. investigated a castor oil-based photopolymer for DLP 3D-printing and demonstrated its reprintability via dissociative bond exchange reactions [47].

It should be noted that most of these systems rely on a radically-induced chain growth reaction of (meth)acrylate moieties, which often yields photopolymer networks that suffer from high network heterogeneities, low toughness, and low monomer conversion [48,49]. The addition of thiols as chain transfer agents does not only improve the network homogeneity but also significantly enhances the toughness and overall mechanical properties of the photopolymers. At higher thiol concentrations, the curing mechanism changes from a chain growth to a step growth reaction (thiol-click chemistry), where high reaction yields have been greatly exploited in photopolymerization systems [50,51,52].

In this work, we study the use of a biochemical-derived acrylated linseed oil (AELO) as a building block for 3D-printable thiol-click networks containing dynamic hydroxyl ester bonds. An organic phosphate ester is added as catalyst, which activates transesterification reactions within the photopolymer network at elevated temperature. Cure kinetics, stress relaxation, and mechanical properties are characterized as a function of the network composition. With a selected formulation, various objects with healing and shape memory properties were fabricated using digital light processing (DLP) 3D-printing. To demonstrate the versatility of our developed materials, we further performed reshaping experiments in the macroscopic range using 3D-printed objects. These could inspire the design of functional devices for numerous industrial fields such as the electronics industry or soft robotics, where high-resolution materials with improved sustainability and additional functionalities are desired.

## 2. Materials and Methods

One hundred-percent bio-based epoxidized linseed oil was purchased from HOBUM OLEO Chemicals (Hamburg, Germany). Ethylene glycol bis-mercaptopropionate (EGMP) and trimethylolpropane tri(3-mercaptopropionate) (TMP3MP) were provided by Bruno Bock Chemische Fabrik (Marschacht, Germany). Acrylic acid, bis(2-methacryloyloxy ethyl) phosphate (DMEP), phenyl bis(2,4,6-trimethylbenzoyl) phosphine oxide (BAPO), and dichloromethane (DCM) were supplied by Sigma-Aldrich and used as received.

### 2.1. Synthesis of Acrylated Linseed Oil

Acrylated linseed oil (AELO) was synthesized by adding epoxidized linseed oil (ELO) (100.05 g), acrylic acid (23.076 g, 0.32 mol), and butylated hydroxytoluene (0.33 g) to a three-necked round-bottomed flask. After stirring the mixture at 60 °C for 20 min, triphenylphosphine (8.12 g, 0.037 mol) was added. The mixture was stirred for another hour at 60 °C. Further acrylic acid (22.055 g, 0.306 mol) was then added, and the mixture was kept at 60 °C overnight under stirring. Subsequently, the mixture was allowed to cool down to room temperature and was dissolved in 100 mL of ethyl acetate. Excess acrylic acid was removed by washing the organic phase three times with saturated sodium bicarbonate solution followed by a washing step with brine. The organic layer was dried over sodium sulfate and the solvent was removed under reduced pressure. The product was obtained as yellow liquid with an acrylate functionality of 3.2.

^1^H-NMR (400 MHz, CDCl_3_): δ [ppm] = 6.52–6.32 (m, H_a_); 6.25–6.01 (m, H_b_); 5.94–5.77 (m, H_c_); 3.20–2.84 (m, H_d_, H_e_); 2.38–2.20 (s, H_f_).

### 2.2. Resin Preparation

AELO (70 wt%), the transesterification catalyst DMEP (8 wt%), the photoinitiator BAPO (2 wt%), and 0.05 wt% Sudan II as photo-absorber were mixed and stirred in an ultrasonic mixer for 30 min at 40 °C until BAPO and Sudan II were dissolved. The photo-absorber (Sudan II) was added to prevent over-polymerization and to ensure high resolution during DLP 3D-printing. The respective thiol crosslinker (20 wt%) was added after cooling the formulation to room temperature. Light-protected glass vials were utilized for the preparation of the resins. The compositions are summarized in Table 1 and the chemical structures of the components in the resin formulations are provided in Figure 1.

### 2.3. Digital Light Processing (DLP) 3D-Printing

DLP 3D-printing was performed on an Anycubic Photon Zero printer (Shenzhen, China) with a LED 405 nm light source. The overall light exposure was controlled using bottom and normal exposure settings with a corresponding duration of 90 s and 70 s, based on the optimal monomer conversions estimated from the FTIR spectroscopy. The 3D-printed structures were built up layer-by-layer with a layer thickness of 50 µm. Building and retracting speed of the platform were set to 2 mm s^−1^. An optical microscope (Olympus BX 51, Tokyo, Japan) was utilized to record images of the DLP 3D-printed test structures with a Color View IIIu digital camera (Soft Imaging System, Münster, Germany).

### 2.4. Evaluation of Reaction Kinetics and Material Characterization

Fourier transform infrared radiation (FTIR) spectroscopy was performed on a Vertex 70 spectrometer (Bruker, Billerica, MA, USA) to analyze the reaction progress over the illumination time. 16 scans were measured for each exposure time interval in transmission mode between 4000 to 700 cm^−1^, with a resolution of 4 cm^−1^. For sample preparation, the liquid resin (1.2 µL) was cast between two CaF_2_ slides. Photocuring reactions were studied by illuminating the resin with a LED light lamp (zgood^®^ wireless) with a power density of 3.3 mW cm^−2^ (λ = 420–450 nm). Reaction conversions were estimated from the normalized characteristic peak areas of acrylate (1635 cm^−1^) and thiol (2570 cm^−1^) groups with the corresponding exposure dose using the software OPUS. Normalization of the acrylate and thiol peaks was carried out using C=O peak (1750 cm^−1^), which remained unaltered during the curing process (Appendix A).

FT-IR spectra of cured and thermally-treated samples (180 °C for 4 h under air) were recorded with a FT-IR spectrometer (Vertex 70, Bruker, Billerica, MA, USA) integrated with a reflection diamond attenuated total reflection (ATR) accessory (Platinum ATR). A total of 16 scans were measured for each sample from 4000 to 400 cm^−1^.

Stress relaxation experiments were carried out at temperatures between 140 and 180 °C on DLP 3D-printed samples (*d* = 10 mm, *w* = 1 mm) with a rheometer (Anton Paar Physica MCR-501, Graz, Austria) in parallel plate configuration. Prior to the relaxation measurements, samples were equilibrated at 20 N normal force and the corresponding temperature for 20 min. Subsequently, 3% strain was applied and the decreasing stress was recorded over a period of time. Tensile properties of the materials were characterized on DLP 3D-printed specimens (*l* = 30 mm, *w* = 10 mm, and *h* = 1.5 mm) with Zwick Roell Z1.0 static material testing equipment (Ulm, Germany) at a cross-head speed of 250 mm min^−1^.

The glass transition temperatures (T_g_) of the DLP 3D-printed samples were measured by differential scanning calorimetry (Perking Elmer DSC 400, Waltham, MA, USA). A heating rate of 20 K min^−1^ was applied from −50 °C to 200 °C under nitrogen atmosphere. The T_g_ of the samples was determined by taking the midpoint of the heat capacity from the second heating run.

The viscosity of the resins was determined by using a compact modular rheometer (MCR 102 Anton Paar, Graz, Austria) with a CP60-0.5/TI cone (49.97 mm diameter and 1.982° opening angle). Each measurement was performed with 1.2 mL resin at 30 °C and a shear rate ranging from 0.1 to 300 s^−1^.

The gel content of the photopolymers was measured by immersing the 3D-printed samples in dichloromethane (DCM) at 20 °C for 48 h. After removing and drying the samples with paper tissues, the specimens were placed in an oven at 40 °C until constant weight. The gel content was determined by taking the mass ratios *m*_dry_/*m*_initial_, where *m*_dry_ represents the weight of the final dried samples and *m*_initial_ represents the weight of the initial samples. Five samples of each resin were measured and the arithmetic average was determined.

Thermal healing experiments were carried out on DLP 3D-printed rectangular bars (*l* = 30 mm, *w* = 10 mm, *h* = 1.5 mm) with a central cavity (*d* = 5 mm). Circular discs (5 × 1.5 mm) and control samples without cavity (same dimensions) were also printed with the resin formulation AELO-EGMP. For evaluating the healing efficiency, the circular discs were placed inside the cavities of the rectangular bars and the samples were thermally treated at 180 °C for 4 h in the oven. Tensile tests were performed on rectangular bars with a cavity, healed samples, and control samples with Zwick Roell Z1.0 static material testing equipment (Ulm, Germany) at a cross-head speed of 250 mm min^−1^.

Shape memory experiments were performed with DLP 3D-printed grippers (*d* = 50 mm, *w* = 1 mm) after a thermal treatment at 180 °C for 4 h. The first shape of a 3D-printed gripper was fixed by heating the sample to 180 °C for 2 h. For programming the second shape, the gripper was cooled down to 30 °C, re-shaped, and further cooled down to −20 °C in a deep freezer. By heating the gripper to 30 °C, the first shape was regained within 30 s. The permanent shape of the specimen was restored by heating the gripper to 180 °C for 2 h.

## 3. Results and Discussions

### 3.1. Synthesis and Network Evolution of Dynamic Thiol-Acrylate Photopolymers

The acrylated linseed oil (AELO) was obtained via the nucleophilic ring opening reaction of an epoxidized linseed oil with acrylic acid (Appendix A), as reported in the literature [53]. The one-pot synthesis afforded the acrylated product at high yields, which was confirmed by ^1^H NMR spectra in deuterated chloroform (Appendix A). The signal at 2.3 ppm is assigned to the three CH_2_ groups (H_f_) attached at the α-position of the ester carbonyl. This peak was also employed as a reference to quantify the introduced acrylates (5.85, 6.15, 6.4 ppm). By setting the integral value of H_f_ to 6, the integrals of H_a-c_ could be directly used to determine the average acrylate functionality per triglyceride molecule. Furthermore, the decrease in epoxide functionality (3.0 ppm) could be observed from 5 to 0.4 per triglyceride molecule, showing a successful reaction. According to the NMR data, an AELO with an acrylate functionality of 3.2 was synthesized. Since the residual epoxide functionality was 0.45, it was calculated that 90% of the epoxide groups participated in the reaction. However, since more epoxides reacted than acrylates formed, it is assumed that a part of the epoxy groups did hydrolyze or participated in other side reactions. An acrylate functionality of 3.2 was sufficient for all further reactions; therefore, no effort was made to achieve higher epoxide conversions.

AELO-thiol resins were prepared by employing two different types of thiols with varying functionality: the trifunctional thiol TMP3MP and the difunctional thiol EGMP. BAPO was used as a radical photoinitiator, whose extended absorption profile allowed for the curing of the AELO-resins under visible light exposure (Figure 2). The curing kinetics of the reaction were monitored by following the depletion of the characteristic stretching bands of S-H bonds at 2570 cm^−1^ and the C=C bond wagging band at 1635 cm^−1^ over the exposure dose (IR spectra prior to and after illumination are shown in Appendix A). In both resins, the final acrylate conversion was significantly higher (87% and 97%) compared to the thiol conversion, which amounted to 46% and 48% for AELO-TMP3MP and AELO-EGMP, respectively. This can be explained by the participation of the acrylate moieties in both reaction pathways—the chain growth homo-polymerization (acrylate-acrylate) as well as the step-growth thiol-ene addition reaction (thiol-acrylate)—leading to a mixed-mode reaction mechanism [26,27,54]. While chain growth polymerization reactions show fast conversions and lead to high molecular weights, the step growth polymerization favors the formation of lower molecular weight species, resulting in delayed gelation, more homogenous networks, and lower shrinkage stress [26,27,50,54,55]. It is reported in the literature that the rate of homo-polymerization is 1.5 times higher than the rate of hydrogen abstraction from thiol moieties [50,56], which ultimately leads to the higher conversion of acrylates in thiol-acrylate-based networks.

The final conversion of acrylate moieties was higher in the case of AELO-EGMP (97%), in contrast to AELO-TMP3MP (87%), while the thiol conversion also decreased with increasing thiol functionality (Figure 3a,b), which might be due to the diffusion limitation of the reactive monomers. The increase in thiol functionality leads to a decrease in gel point conversion and the polymerization rate slows down due to diffusional constraints. Moreover, almost similar reaction kinetics were recorded during the first 70 s of illumination, and acrylate conversion was above 70% in both cases. It should also be noted that the trifunctional thiol acts as a crosslinker, whereas the difunctional thiol acts as a chain extender and increases the molecular weight between adjacent crosslinks [27,55,57]. This was confirmed by additional sol-gel analysis. The gel content of the cured samples was determined by immersing the 3D-printed samples for 48 h in dichloromethane, and the residual weight of the dried samples was recorded. The networks formed with difunctional thiol (EGMP) exhibited a gel content of 71.5% ± 1.9%, while the networks formed with trifunctional thiol (TMP3MP) had a gel content of 77.7% ± 2.3%. The higher gel content of the AELO-TMP3MP system can be explained by an increased crosslink density due to the higher functionality of the thiol used. Generally, the low gel content can be related to an incomplete monomer conversion and, especially, the thiol crosslinkers, which were extracted from the networks during swelling.

### 3.2. Network Properties of Dynamic Thiol-Acrylate Photopolymers

The residual thiols act as chain lubricants and in combination with flexible thioether bonds, and the mobility of the networks is significantly enhanced, which leads to glass transition temperatures (T_g_) of −18 °C and −10 °C for AELO-EGMP and AELO-TMP3MP, respectively. The related DSC curves are shown in Appendix A. For uniaxial tensile tests, samples were prepared via DLP 3D-printing. Based on the low T_g_, the networks were soft at room temperature and were characterized by a low tensile strength, which did not exceed 1.18 MPa and 0.58 MPa for TMP3MP- and EGMP-based formulations, respectively. The ultimate elongation of AELO-EGMP (19 %) was lower compared to AELO-TMP3MP (38 %) (Figure 3c). By thermal annealing of the 3D-printed samples at 180 °C for 4 h, the Young’s modulus and the mechanical properties were improved (Figure 3d). After the thermal treatment, both networks underwent an increase in the tensile strength, and exhibited a maximum strain of 31 % (AELO-EGMP) and 37 % (AELO-TMP3MP), and a breaking stress of 1.0 MPa (AELO-EGMP) and 1.3 MPa (AELO-TMP3MP), respectively. This improvement in toughness and modulus can be attributed to the thermally-induced rearrangements and the reduction in shrinkage stress. Previous work demonstrated that the thermal treatment also enables the formation of hydrogen bonds, which further increases the tensile properties [27,58,59]. The analysis of ATR FTIR spectra of the thermally-treated samples also clearly indicates the depletion in the -OH groups’ region (3300–3600 cm^−1^), confirming the formation of hydrogen bonds (Appendix A).

The kinetics of the thermo-activated bond exchange reactions was characterized by stress relaxation experiments between 140 °C and 180 °C. Samples with a diameter of 10 mm were DLP 3D-printed and the impact of the thiol crosslinker on the exchange kinetics was determined. Based on the Maxwell model, the characteristic time for stress relaxation τ* can be determined from the measured relaxation plots when the normalized relaxation modulus (G_t_/G_o_) reaches 37 % (1/e) of its initial value [60]. Generally, the relaxation time (τ*) in vitrimers follows an Arrhenius type temperature dependency τ* = τ_o_ exp (E_a_/RT), where E_a_ represents the activation energy, T the temperature, and R the general gas constant [60,61,62].

Figure 4a represents a time-dependent evolution of the relaxation modulus at 180 °C for AELO-TMP3MP, AELO-EGMP, and AELO-EGMP without transesterification catalyst. The presence of DMEP as a transesterification catalyst results in stress relaxation of the networks, albeit at different rates. Furthermore, the thiol-acrylate networks were able to relax stress in the absence of a catalyst; however, the rate was much lower. This relaxation is contributed by the thermal release of volumetric shrinkage stresses generated during the network formation [26,63]. Without the catalyst, AELO-EGMP reached 37% stress within a time period of 384 min at 180 °C (Figure 4a), whilst the catalyzed AELO-EGMP network exhibited a τ* of 22 min.

The relaxation rate of catalyzed AELO-EGMP (22 min) is much faster in comparison to catalyzed AELO-TMP3MP (77 min), which can be attributed to the higher crosslinking degree of AELO-TMP3MP, as the number of -OH and ester groups was comparable in both networks. Moreover, the stress-relaxation of both networks shows a clear temperature dependency and the bond exchange rate increases with rising temperature (Figure 4b,c). In Figure 4d, τ* is plotted versus 1/T in a semilogarithmic scale and the linear trend obtained for both networks confirms the Arrhenius type temperature dependency and the vitrimeric behavior of the developed networks [64]. The E_a_ of the networks was also determined, by taking the slope (m = E_a_/R) of the straight line fitted to the data, and it amounted to 77.58 and 79.88 kJ·mol^−1^·K^−1^ for AELO-EGMP and AELP-TMP3MP, respectively.

### 3.3. Thermally Triggered Healing and Shape Memory Properties of DLP 3D-Printed Objects

Due to the lower viscosity of AELO-EGMP resin (2.1 Pa·s obtained at a shear rate between 0.1–300 s^−1^) compared to AELO-TMP3MP (11.9 Pa·s obtained at the same shear rate range), it was further applied to DLP 3D-print objects to study the material resolution, thermal healing, and shape-memory performance.

A comb-like structure was DLP 3D-printed with holes of different sizes to monitor the printing capabilities (Figure 5a). We observed that it was possible to print holes with a diameter up to 250 µm. At lower feature sizes, the material started to lose resolution and no open holes could be obtained (Figure 5b).

Due to the dynamic nature of the developed thiol-acrylate networks, the AELO-EGMP photopolymer exhibits triple shape memory and the ability to undergo controlled and macroscopic deformation when the sample is heated and programmed beyond its two transition temperatures, T_v_ and T_g_. As presented in Figure 6a,b, the permanent shape of a DLP 3D-printed and thermally-annealed flower was altered by heating it above the highest transition temperature (T_v_), along with applying an external deformation force. A temperature of 180 °C was applied since rheological experiments revealed a fast stress relaxation at this temperature (Figure 4b). The first temporary shape was fixed by cooling the sample to 30 °C, which is well above the T_g_ of the material and enabled the second programming. By applying a deformation force and cooling the sample below T_g_ (−20 °C), the second temporary shape was fixed. Subsequently, the specimens were able to sequentially recover the different shapes under thermal stimulus without applying any force. A recovery time of 30 s was recorded for the first temporary shape, which also highlights the potential application of partly bio-based thiol-acrylate resins in biomedical and soft robotic systems.

In a second step, the thermally-induced healing and repair abilities of the AELO-EGMP photopolymer have been studied. Rectangular bar-shaped samples with (defect) and without (defect-free control sample) a circular cavity in the center were DLP 3D-printed. A disc fitting in the cavity was further printed and inserted in the defect sample. For the thermal healing/welding, the sample was placed at 180 °C for 4 h in the oven without applying any pressure (Figure 6c). Subsequent tensile testing of the healed samples revealed that the stress–strain curves are nearly identical to the defect-free control sample, giving rise to the high healing efficiency of the AELO-EGMP photopolymer (Figure 6d).

## 4. Conclusions

In the present work, acrylated linseed oil was synthetized and used as a building block for the preparation of dynamic thiol-acrylate photopolymers, which relied on thermo-activated transesterification. Structure–property relationships with two different thiols, varying in their functionality (bi- and tri-functional), were comprehensively described. The bifunctional thiol yielded a lower crosslinked photopolymer with a T_g_ of −18 °C compared to the trifunctional counterpart (T_g_ = −10 °C). The lower crosslinking degree enhanced the mobility of the AELO-EGMP networks, which accelerated the bond exchange reactions at an elevated temperature (>140 °C). With both thiols, thiol-acrylate networks were rapidly formed upon visible light exposure, which is crucial for DLP 3D-printing. Taking advantage of the lower viscosity, EGMP-AELO was applied in subsequent printing experiments and it was found that objects with a feature size up to 550 µm could be conveniently fabricated. The EGMP-AELO network comprised ample free ester and hydroxyl groups, which, in the presence of an organic phosphate ester as a catalyst, underwent bond exchange reactions at elevated temperatures (>140 °C). The related macroscopic reflow was used for the healing and re-shaping of the printed structures. Tensile properties of defect free samples could be nearly fully recovered after a thermal treatment at 180 °C for 4 h. Furthermore, the 3D-printed structures were able to undergo triple-shape memory and two different shapes were programmed by exploiting the topological freezing temperature and the glass transition temperature of the dynamic thiol-acrylate photopolymers.

## Figures and Tables

**Figure 1 polymers-14-05377-f001:**
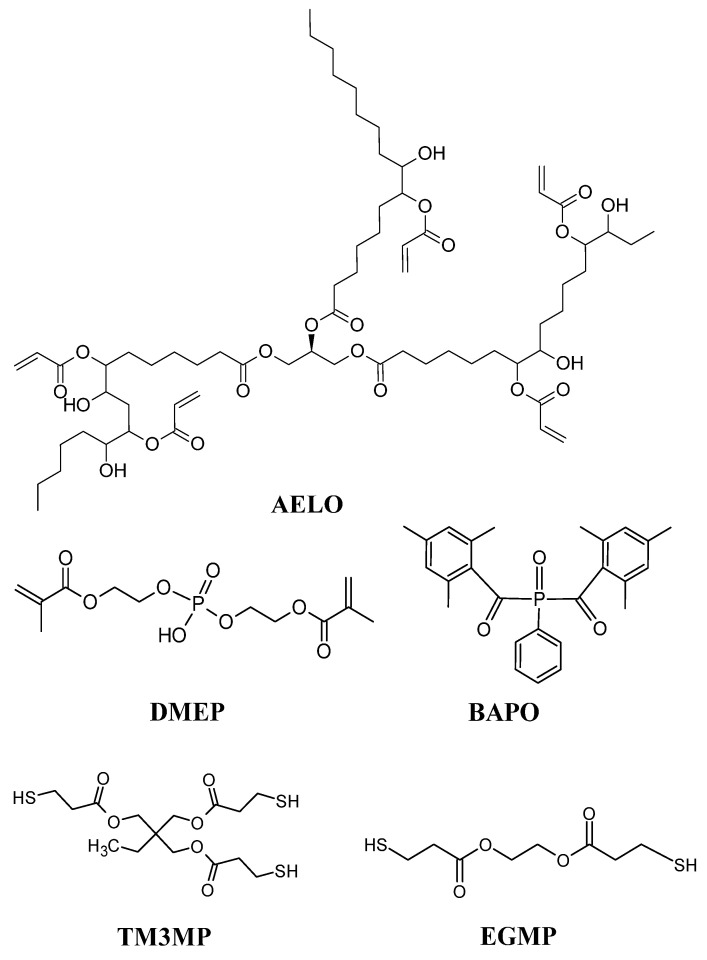
Chemical structures of the components used in the resin formulations.

**Figure 2 polymers-14-05377-f002:**
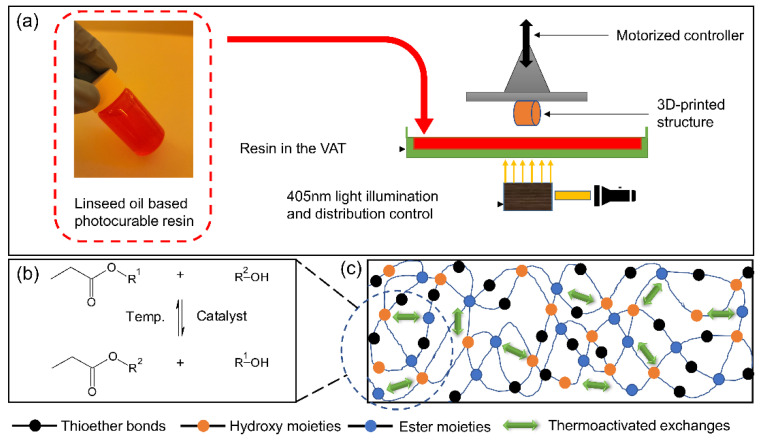
(**a**) Schematics of the digital light processing (DLP) 3D-printing for the printing of bio-based resins, the inset figure shows the actual resin formulation developed for DLP 3D-printing; (**b**) thermoactivated transesterification with the application of temperature and catalyst; (**c**) conceptual diagram of the cured network showing permanent thioether bonds and thermally-activated hydroxy-ester exchange reactions.

**Figure 3 polymers-14-05377-f003:**
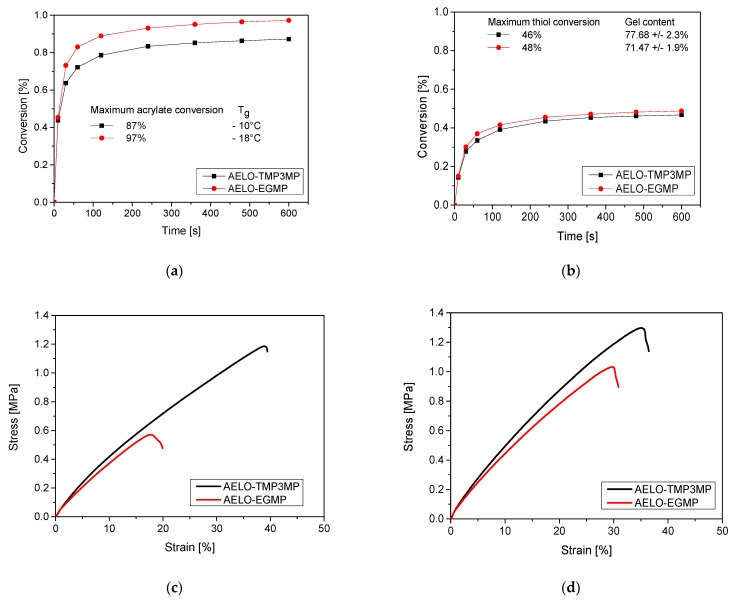
Monitoring of the conversion of (**a**) acrylate (1635 cm^−1^) and (**b**) thiol (2520 cm^−1^) groups over illumination time (405 nm, 3.3 mW cm^−2^) via FTIR experiments. Stress–strain curves of AELO-EGMP and AELO-TMP3MP photopolymers (**c**) prior to and (**d**) after thermal treatment (180 °C, 4 h).

**Figure 4 polymers-14-05377-f004:**
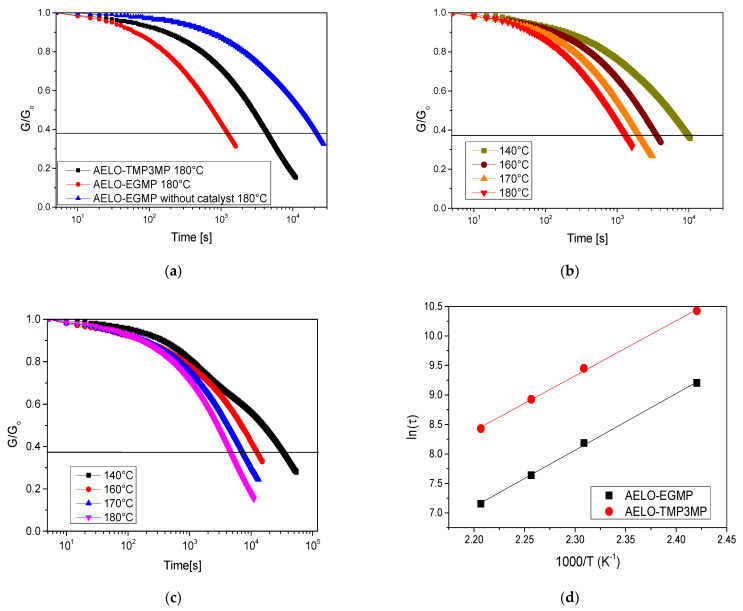
(**a**) Time-dependent relaxation modulus data recorded at 180 °C for AELO-TMP3MP and AELO-EGMP networks. Time-dependent relaxation modulus of (**b**) AELO-EGMP and (**c**) AELO-TMP3MP networks recorded at a temperature range between 140 and 180 °C. (**d**) Arrhenius plots of AELO-EGMP and AELO-TMP3MP confirming the linear dependency, which is characteristic for dynamic polymer networks following an associative bond exchange mechanism.

**Figure 5 polymers-14-05377-f005:**
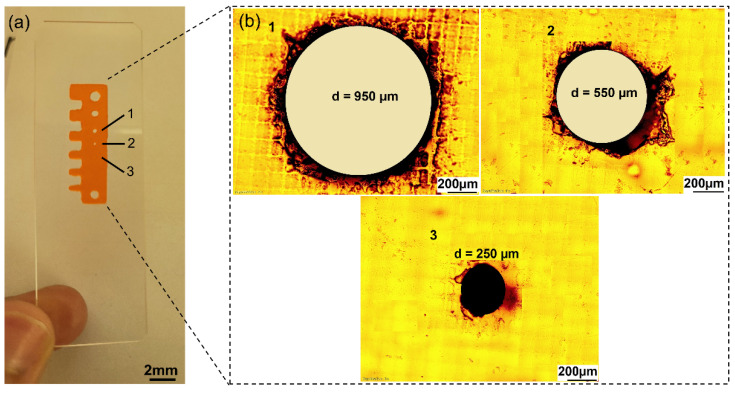
(**a**) Resolution test using a DLP 3D-printed comb structure. (**b**) Analysis of the circular cavities under the microscope for determination of the maximum printing resolution.

**Figure 6 polymers-14-05377-f006:**
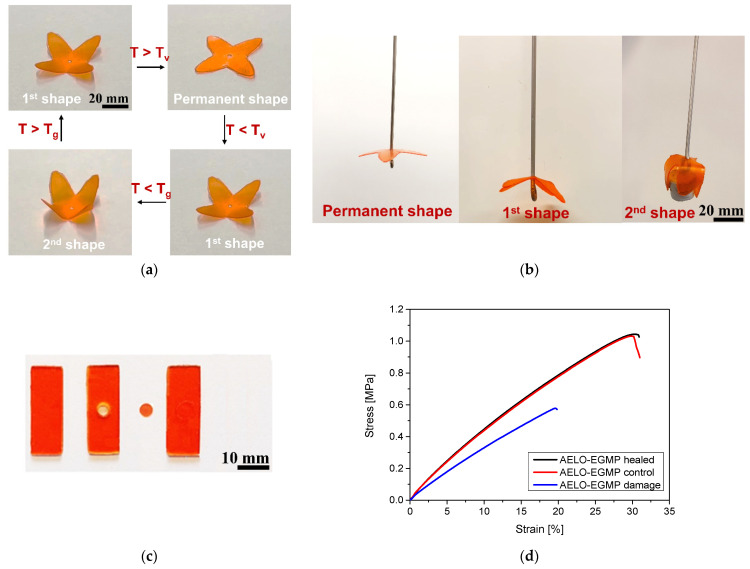
(**a**) Photographs representing the triple shape memory of DLP 3D-printed AELO-EGMP structures. The first shape was programmed above the T_v_ of AELO-EGMP while the second shape of material was programmed using the T_g_ of the photopolymer. Heating of the programmed object sequentially recovered the permanent shape of the 3D-printed flower; (**b**) DLP 3D-printed structure representing the potential application as gripper by employing shape memory properties; (**c**) DLP 3D-printed control sample, damaged sample with a circular cavity, the disc, and the healed sample for analysis of thermal welding and healing properties; (**d**) Stress–strain curves of control, damaged, and healed samples. Thermal healing was performed at 180 °C for 4 h.

**Table 1 polymers-14-05377-t001:** Composition of AELO-thiol resins used in this study. Formulations contained 8 wt% DMEP, 2 wt% BAPO, and 0.05 wt% Sudan II.

Formulation ID	AELO (wt%)	Thiol (wt%)
AELO-EGMP	AELO (70)	EGMP (20)
AELO-TMP3MP	AELO (70)	TMP3MP (20)

## Data Availability

Not applicable.

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
