# Peer review of "Vat Photopolymerization 3D-Printing of Dynamic Thiol-Acrylate Photopolymers Using Bio-Derived Building Blocks"

_polymers, 2022, doi:10.3390/polym14245377_

Round 1

Reviewer 1 Report

The manuscript presents the synthesis of dynamic thiol-acrylate photopolymers for the digital light processing 3D printing. These photopolymer compositions have advanced properties and demonstrate capability for healing and re-shaping the 3D-printed objects by exploiting the topological freezing temperature and the glass transition temperature of the photopolymers. Two compositions of AELO-thiol resins have been developed and their properties such as cure kinetics, stress relaxation and mechanical properties have been characterised. Furthermore, the capability of the new materials for the digital light processing 3D printing of objects with a structure size of 550 µm and their triple-shape memory have been demonstrated.

The manuscript is well written, has a comprehensive introduction to the state-of-art, clearly defines the aim of the current research, provides a broad discussion of the experimental results and describes the conclusions in accordance with the obtained results.

I was pleased to review such a high quality manuscript and would like to congratulate the Authors on the interesting paper. There are few points that I would like to clarify.

1.       What material modifications can be done to improve the resolution of the material for 3D printing?

As seen from Figure 5, the shape of the hole is far from the perfect circle.

2.       Is this material suitable for producing cavities with more complex shapes such as a spiral, for example?

3.       Is this material suitable for functionalisation with nanoparticles to enhance its toughness further, for example?

Author Response

(1) What material modifications can be done to improve the resolution of the material for 3D printing? As seen from Figure 5, the shape of the hole is far from the perfect circle.

Answer: To further improve the resolution of the material, reactive diluents could be added to reduce the viscosity of the resin while increasing its reactivity. In this context, a linseed oil-based acrylate with lower functionalities could be used to reduce the viscosity of the system.

(2) Is this material suitable for producing cavities with more complex shapes such as a spiral, for example?

Answer: Yes, it is also suitable for the production of more complex shapes. The factor, which needs to be considered for printing complex indentations or patterns would be the size of the structures. If the size of the structure or cavities is very small (i.e. < 550 µm), reactive diluents need to be introduced in the resin formulation to improve the printing resolution as mentioned above.

(3) Is this material suitable for functionalisation with nanoparticles to enhance its toughness further, for example?

Answer: Yes, the material is also suitable for functionalisation with nanoparticles. Thank you for the input!

Reviewer 2 Report

In the manuscript entitled "Vat photopolymerization 3D-printing of dynamic thiol-acrylate photopolymers using bio-derived building blocks” authors studied the possible improvement of the  environmental footprint of 3D-printed objects, by combining the hydroxyl ester links, undergoing a catalyzed transesterification at elevated temperature, with an acrylate monomer derived from renewable resources.

The topic is interesting and relevant to the field due to the fact that 3D printing technology is nowadays widely used and shows a great potential for various industrial fields. Furthermore, a major drawback of photopolymers is their lack in recyclability and repairability as liquid resin transforms to a solid covalently crosslinked polymer network during the printing process. By introducing a biochemical derived acrylated linseed oil as a building block for 3D-printable thiol-click networks containing dynamic hydroxyl ester bonds a great improvement in the environmental aspect has been described.

The manuscript is well organized, the authors used the scientific methods, and they are adequately described. The results are clearly presented and reproducible based on the details of the methods presented. In the conclusion part authors presented the findings of the properties of the dynamic thiol-acrylate photopolymers which demonstrate promising healing, shape memory and re-shaping properties offering a great potential in 3D printing.

The references cited in the manuscript are recent, mostly within the last 10 years. The figures, tables, images, and schematics are presented appropriately and clearly. Data presented in charts are properly presented and easy to interpret and understand.

The following corrections should be made:

- the abbreviation pKa should be described (line 63)

- references should be written as directed.

Acceptance of the manuscript is suggested with those mentioned corrections.

Author Response

(1) The abbreviation pKa should be described (line 63)

Answer: The pKa- value describes the negative logarithm of the acid dissociation constant (Ka) in solution. As suggested, we added the explanation in the revised manuscript (page 2, line 63) and highlighted the changes in yellow.

(2) References should be written as directed.

Answer: We checked the references carefully and changed them in accordance to the journal’s policies.